# The Properties of a Ship’s Compass in the Context of Ship Manoeuvrability

**DOI:** 10.3390/s23031254

**Published:** 2023-01-21

**Authors:** Andrzej Felski, Krzysztof Jaskólski

**Affiliations:** Department of Navigation and Maritime Hydrography, Polish Naval Academy, ul. Śmidowicza 69, 81-127 Gdynia, Poland

**Keywords:** attitude and heading reference systems, ship’s manoeuvrability, heading errors, spectrum

## Abstract

In evaluating the accuracy of most navigation measuring systems, it is accepted, as a rule, that measurement errors are characterised by a normal distribution. With reference to the compass, the approach of most producers is similar. However, in the case of this measuring device, the dynamics of the ship should also be taken into account. The problem is that any changes in the ship’s heading can be measured exclusively with the use of a compass. Until quite recently, this device was built based on mechanical elements, so it possessed its own dynamic properties. This means the appearance of specific, positive feedback (self-reinforcing feedback) because if the compass did not point to the correct heading, it could lead the ship to stray from the correct heading. On the other hand, it could mean an incorrect compass setup, even though the ship had the correct heading. Any incorrect indications of the compass were then interpreted as a ship’s departure from the correct heading. This problem was not essential in the era of magnetic compasses because the errors in these compasses are relatively constant, unlike the errors in gyrocompasses, which have an oscillatory and random character and, thus, it is not possible to describe them accurately with mathematical relations. This issue was already perceived before WWII, when the Anschutz Company proposed, among other solutions, using the so-called Schuler period in the construction of gyrocompasses. Fibre optic gyrocompasses do not possess mechanical sensors, so the variability of their indications is of a different character. However, computational processes, as well as applied inertial sensors, also cause certain errors of an oscillatory nature. This raises the following questions: what is the spectrum of the error frequency of such compasses, and what is the influence of the ship’s movement on them? The authors attempted to evaluate this phenomenon by performing measurements made on board three hydrographic platforms and comparing them with the headings indicated by other compasses.

## 1. Introduction

Modern navigation is dominated by onboard electronics, which are clearly displayed by multiple monitors on the bridge of a typical modern vessel. This picture is reinforced by the widespread reliance on a Global Navigation Satellite System (GNSS)’s satellite signals. This, in turn, makes the problem (which, twenty years ago, was of paramount importance to the navigator) of answering the question “where am I positioned now?” a trivial issue. On a typical vessel, a GNSS system (usually a GPS receiver) provides new, up-to-date geographical coordinates of the vessel every second, usually with an accuracy of no less than 5 m. All this means that the issue of the safe passage of a vessel is sometimes interpreted as something that should not cause difficulties, especially for the generation that has become accustomed to computers, smartphones, and similar devices since birth.

However, it should be borne in mind that modern vessels are colossal, with lengths of up to 400 m, that no other man-made mobile objects can compare with. Their enormous weights sometimes exceed 200,000 tonnes, while their specific design results in poor manoeuvrability with enormous inertia. Although people have been building ships for centuries, research into the manoeuvrability of vessels actually began as late as the 1930s. Although the first model studies were initiated by Tideman and Froud as early as the 19th century, it is the name of Gunter Kempf that should be primarily mentioned in the context of investigating such features of a vessel as its ability to alter course (manoeuvrability) and maintain course spontaneously (directional stability). In 1932, he proposed a zig-zag test [1], i.e., a ship’s stability test procedure, which is still in use today. The procedure involves deflecting the rudder cyclically by a specific value, and when the vessel’s course changes by that predetermined value, the rudder is deflected to the opposite side by the same value. One of the more popular variants of this test is the 20-20 variant, although other variants are used as well. For example, for large vessels, the 10-10 or 10-20 variant is preferred (where the former number represents the rudder angle, and the latter represents the course change). This test is usually continued until five subsequent rudder deflections have been performed, and its essence is primarily to determine the vessel’s yawing period t_m_.(see Figure 1) [1].

Based on the data obtained from this test, the dimensionless directional stability coefficient is calculated.
(1)$$E=V_{o}\;\left(t_{m}/L_{p}\right)$$
where

E is the dimensionless directional stability coefficient;

$V_{o}$ is the initial velocity;

$t_{m}$ is the yawing (course oscillation) period;

$L_{p}$ is the vessel’s waterline length (the length between the perpendiculars).

Such information provides a synthetic indicator of the extent to which the vessel has a tendency to maintain a stable course. The higher the E value, the more stable the vessel is, but the more difficult it is to make it turn. A kind of alternative to this test is the measurement of the vessel’s manoeuvrability, i.e., the ability to alter course quickly. This feature is described by the manoeuvrability coefficient calculated according to the relationship (2).
(2)$$K=D_{u}/L_{p}$$
where

*K* is the dimensionless manoeuvrability coefficient;

*D_u_* is the fixed circulation diameter;

*L_p_* is calculated as in Equation (1).

In practice, the vessel’s crew is more interested in the second information, in particular the circulation diameter, as knowledge of this parameter is necessary to keep the vessel on a fairway that requires course changes. The K/E ratio is referred to as the Nomoto number [2], which, in generalised terms, represents the manoeuvrability of the vessel and is important for autopilot programming. For a typical transport vessel, the Nomoto number oscillates around a value of 0.5. Currently, other relationships [3] are also known and used, which enable an assessment of how stable a vessel is on course and how manoeuvrable it is, although this issue is not the subject of this article. It has also been addressed as an example in order to explain the nature of the issue and how it relates to the compass characteristics under consideration.

Such considerations usually disregard the possibility of orienting the vessel in the right direction (i.e., stabilising its course), which requires the presence of any compass. This issue can be considered irrelevant when considering the passage of a vessel through the ocean or open sea, where the distances to the bottom can be measured in kilometres or at least hundreds of metres, and the nearest natural or man-made objects are found at distances of tens of kilometres. However, in this case, it is desirable to keep the vessel on course more accurately, as this translates into fuel and time savings. This issue, however, becomes particularly relevant in water areas where precise course maintenance is critical, e.g., on waterways and fairways. We are very often informed about incidents when a vessel has blocked a navigation channel, encountered underwater obstacles, or collided with other sea users. One of the reasons for such incidents is the vessel’s directional stability and manoeuvrability, but the quality of the compass, which is fundamental to carrying out safe manoeuvres, cannot be overlooked.

Controlling a vessel, particularly a large one, requires the transmission of information on its orientation in relation to the geographical direction with a high degree of accuracy. The documents relating to the most commonly used gyrocompasses [4,5] define these requirements as a function of latitude, and at latitudes of 60°, this error cannot be greater than 1.5°, while on the equator, it cannot be greater than 0.75°. The requirements for other solutions do not deviate substantially from these values. This article addresses the accuracy of compasses on vessels and drones designed for hydrographic surveys. Such operations are characterised by their specificity because, on a typical vessel, the primary aim is to stabilise the course, and only in exceptional cases does the crew attempt to keep the vessel on the preset route. This is the case when navigating on waterways, fairways, and recommended routes, etc. Hydrographic surveys require the survey platform, or, more precisely, its centre of gravity, to be maintained above the planned survey line. In such a case, since the (human or automated) helmsman seeks to minimise the value of the deviation from the planned route (Cross Track Error—XTE, see Figure 2a), changes to the course will be determined by a different factor than when on an average merchant ship. On a typical vessel, the aim is to minimise deviations from the course, while on a hydrographic vessel, one can expect greater changes to the course in view of the strategy for minimising XTE and striving for a faster return to the planned route. This results in the movement of such a vessel’s centre of gravity resembling not a smooth sinusoid but a curve, which diagram is presented in Figure 2b.

Gyrocompasses—proposed independently by Anschutz and Sperry at the beginning of the 20th century and until recently built using only a mechanical gyroscope—have been commonly used on vessels for more than 100 years [6]. A gyrocompass is an electromechanical device whose sensor uses the directional properties of a mechanical gyroscope and a pendulum. The latter factor brings about oscillations in its indications, which have several causes. One of these is the accelerations occurring on the vessel (motions of the hull, changes in its velocity vector caused by environmental influences, or manoeuvres resulting from the decisions of the crew). However, one should also realise that all systems using the gyroscopic phenomenon do not work out the direction in relation to the Earth but to an inertial coordinate system. Meanwhile, the navigator is interested in the direction relative to the Earth’s pole. Since the Earth rotates, this direction is also constantly changing if it is related to an inertial reference system. This requires controlling the gyro sensor according to the latitude at which the vessel is located.

However, both the human controlling the vessel and the automaton performing this task can only control based on the compass indications. Therefore, if oscillations occur in the compass indications, a question arises as to the effect of these oscillations on the vessel’s movement. It should also be noted that the oscillatory nature of changes to the vessel’s course can also induce oscillations in the compass indications. Therefore, a kind of positive feedback loop emerges in this system, as it cannot be ruled out that where the compass indicates a change to a course, this could also be the result of compass oscillations and not changes to the vessel’s course.

Classic gyrocompasses are increasingly being replaced by compasses that have no electromechanical components and are most often equipped with sensors using the Sagnac effect (most commonly Fibre Optic Gyro—FOG, and less commonly Ring Laser Gyro—RLG). This particularly applies to vessels on which very precise control is required or situations where the causes of the above-mentioned oscillations are particularly great. Until recently, this referred to warships or specific vessels that are very fast or cover short routes.

Fibre optic gyroscope (FOG) is based on the Sagnac effect discovered at the beginning of XX century. It manifests itself such that light travelling along a closed ring path in opposite directions allows one to detect rotation with respect to inertial space. If the sensor rotates, one path lengthens when counterrotating one is less during the transit time of the light, which can be observed as the interference. The effect is extremely weak, but it can be increased with recirculation in the resonant cavity, which was first used in the so-called Ring Laser Gyros. Taking advantage of the development of optical fibre communication technologies, in mid 1970s, the idea to replace resonant cavity with the fibre coil opened the way to the fibre optic gyroscope as a new, fully solid-state rotation sensor. It was firstly seen as dedicated to medium-grade applications, but today, it reaches strategic-grade performance and surpasses its earlier established competitor. At present, a FOG coil of 10^4^ loops of 10 cm diameter (i.e., 3 km long) has the potential to be use in systems for measuring the movement and turns of almost any platforms, as its long-term bias stability can be below 10^−3^ (°/h). In the marine technology, this appears in a wide spectrum of Attitude and Heading Reference Systems (AHRS) that are universally applied on military platforms as well as on devices which are used for measuring, oceanography, offshore, and other tasks.

Even though the mechanism of action in a FOG seems simple, many technological difficulties limit its reach for high accuracy. This is well described in [7]. An example of this is that, in the case of FOG, a single-mode fibre has actually two orthogonal polarisation modes which propagate with slightly different velocities because of fibre birefringence; thus, additional polarisers and filters are necessary in the scheme. A FOG is composed of the following components:(a)Light source, usually based on erbium-dopped fibre amplifiers;(b)Hundreds of metres of polarisation-preserving fibre coil;(c)An integrated optic circuit with electrodes to generate phase modulation and can provide excellent polarisation selectivity;(d)A fibre coupler to send to the detector light returning from the common input–output port of the interferometer;(e)A digital electronic element that generates the phase modulation and the phase feed-back through a digital to the analog convertor.

The development of this technology results in its increasingly widespread use on other vessels, with particularly rapid development concerning unmanned surface vehicles that are increasingly used at sea. The analytical nature of the operation of these devices suggests avoiding the aforementioned oscillations which have resulted from inertia and are typical for components with a certain weight and dynamics, since the information they contain about the course is a result of calculations and not a result of the dynamics of the sensor being aligned in the meridian plane. However, a question arises as to whether the sequential algorithms used in such systems completely eliminate these problems.

The requirement to precisely control the vessel along the trajectory is particularly important in the context of the wide spectrum of survey tasks related to monitoring, exploration, and many other kinds of research. Many of these tasks are carried out by small vessels, but they are increasingly being replaced by unmanned surface vehicles (USV) and unmanned underwater vehicles (UUV). Such vehicles are generally much smaller than most ships and, consequently, they have different manoeuvring properties. On such vehicles, since a traditional gyrocompass cannot be installed due to its dimensions, it is common to implement devices that perform the functions of a compass and are built on the basis of FOG or, less commonly, RLG. Increasingly, these are also the so-called satellite compasses which are very often integrated with FOG gyroscopes, and even often with other ones; these compasses are built using the MEMS technology. Therefore, a question arises as to what extent the dynamic properties of these devices differ from the properties of traditional compasses and how much they correspond to the specificity of the platforms on which they are to be used.

## 2. Compass Errors and Their Investigation

As far as classic gyroscopes are concerned, the phenomenon of self-oscillation has been known for a long time. This issue is partly addressed in the design of these instruments, a particularly glaring example of which is the consideration of the so-called Schuler period. Moreover, an average merchant vessel travels at a relatively stable velocity and does not change course very often. Gyroscopic compasses, which are built with such operating conditions in mind, perform well on typical sea-going vessels. However, the aforementioned design principles result in such devices exhibiting a very long period of self-oscillation. This manifests itself especially during the launch when the compass setting process lasts for many hours. This is not a great inconvenience when considering that, on an average merchant vessel, this device is not even switched off for several years. However, the above-mentioned oscillations affect the accuracy of compasses on intensely manoeuvring vessels, which is a natural phenomenon on short routes, and on hydrographic vessels, whose specificity involves covering parallel, albeit not very long, survey sections (legs). Following an abrupt change in course or velocity, the course errors can have values that considerably exceed the permissible values, and due to the long period of these oscillations, they should be considered within 30 min of completing a manoeuvre. According to [4], an error caused by an abrupt 180° turn at a velocity of 20 knots, should not exceed ±3° secant (φ), which means that at latitudes close to 60°, this error can exceed 4°. Hydrographic vessels do not operate at such velocities, as their values are rather closer to 7–9 knots. However, the stabilisation period for these devices is of paramount importance here, as it can happen that the time to cover a single section is slightly longer. It seems obvious that the requirements for a ship’s compasses that are specified in the above-mentioned document have been defined taking into account the navigating conditions of a transport vessel. In these considerations, we will omit the fact that the permissible error of information transmission to receivers (no greater than 0.2°) and the permissible static error (no greater than 1°) are indicated there. Of greater importance in practice is the requirement that the dynamic error amplitude should not exceed 1.5° at a frequency below 0.033 Hz. This means that the period of such oscillations should be no less than 30 s. This frequency is quite characteristic of typical merchant vessels, and it can be assumed that the intention is to ensure that the oscillations resulting from the yawing do not coincide with the self-oscillation of the compass in order to prevent the system from being activated. It can be assumed that it is adequate for the properties of sea-going hydrographic vessels. However, it is not clear how this relates to the small manned and unmanned vehicles that are increasingly used for hydrographic work performed in the coastal zone.

These commonly respected standards, and thus the technical data provided by the manufacturers, are limited to defining the mean values of the errors regarded as random events. The phenomenon of the oscillation occurrence in this process is considered to be marginal, with only the maximum frequency value being provided. However, this is actually a random process that should be regarded as a process characterised by a certain frequency spectrum, and the assumption of a normal error distribution is, in this case, questionable. These issues have been the subject of numerous studies [8,9,10,11]. Studies conducted in the time domain are predominant. The frequency analysis has been used as well, inter alia, in previous publications [12,13]. The tests of course accuracy for a small surface vessel have also been studied [14]. Subsequent analyses are based on tests that were performed on a satellite compass PwrPak 7, produced by Canadian company Novatel mounted on a very small surface drone [, ]. The small dimensions of the platform used resulted in a small distance between the antennas of the device, which directly affects the measurement accuracy. However, the author declares an accuracy of 0.07° under static conditions. A comparison with the parallelly recorded Course Over Ground allows the author to conclude that the differences in relation to the satellite compass indications have not exceeded 0.1°. However, tests on the same compass under static conditions were referred to in the study [13], where the mean error, also with a distance of 1.2 m between the antennas, was estimated to be 0.2°. The difference is probably related to the duration of the experiment, as the one described in [14] lasted for 180 min, while the experiment in [13] lasted for 12 h. With regard to satellite compasses, the accuracy of measurements is, to some extent, affected by the satellite configuration.

Testing compass accuracy requires the reference system to be defined. This is rather easy under static conditions, but on a vessel, it is usually difficult to define such a system or indicate a reference system. The first experiments [15] on the possibility of building a compass based on a satellite system receiver boiled down to a simple comparison between the proposed device and the gyrocompass being part of the equipment of the vessel. One study [9] proposed a set of geodetic GPS-RTK receivers as a reference device for the Altitude Heading Reference System (AHRS) under study. As late as the 21st century, systems have emerged that ensure accuracy far superior to that offered by compasses. However, these are still combinations of satellite receivers supported by specific inertial solutions. An example of such a solution is the POS MV OceanMaster system manufactured by USA company Applanix-Trimble [16], which provides accurate information on the vessel’s position, attitude, heading, and velocity. The source of this data is a specific, dual-antenna satellite system solution integrated with excellent sensors for rotational speeds and linear accelerations. The variant used in this case, according to the manufacturer’s declaration, achieved a course error of no more than 0.2°. Unfortunately, this solution was only available on one of the three vessels considered in the article, i.e., the survey ship ORP Arctowski, while the other ones were offered no such opportunity.

The primary aim of this article is to discuss various considerations that can affect the adequacy of a FOG-based compass in relation to the specifics of the hydrographic platform on which it is mounted. Controlling a vessel that carries out hydrographic surveys is very specific, as the main task is to maintain the vessel along the preset line above the bottom, and this is determined by the manoeuvring properties of the vessel, the system for controlling it, and the source of information for this system, i.e., not only the course but also information on how far the vessel deviates from the preset line. Therefore, further considerations refer to the recordings taken on vessels differing in design: a sea-going vessel, an offshore hydrographic catamaran, and an unmanned surface vehicle. These vessels were equipped with a variety of compasses, including a classic gyrocompass, a FOG gyrocompass, and a satellite gyrocompass.

## 3. Results

This study used surveys recorded on three platforms used for statutory hydrographic surveys. Their manoeuvring characteristics differed dramatically, and these included a hydrographic warship (ORP Arctowski), a double-hulled hydrographic motorboat, and an unmanned surface vehicle (Figure 3). All the surveys were carried out in the course of these vessels performing their statutory tasks. The data recorded on the ORP Arctowski survey ship are very similar to those previously used in an earlier publication [17], although, at that time, the recordings were made on a different vessel.

The ORP Arctowski ship is a typical survey vessel of project 874 (in the **NATO** code: Modified **Finik**) with a displacement of 1145 t, a length of 62 m, a width of 10 m, and a draught of 3 m. The Wildcat 40 catamaran is a vessel made of fibreglass-reinforced laminate and is used as a motorboat for carrying out surveys in the offshore zone, with a displacement of 15, a length of 12.9 m, a width of 4.8 m, and a draught of less than 1.3 m. The DriX drone, on the other hand, is an innovative vessel with a length of 7.7 m that resembles a small submarine in shape; it is regarded as a surface vehicle as it operates partially submerged. A characteristic of this vessel is its pod that contains hydroacoustic system sensors submerged to a depth of approx. 2 m and a kind of superstructure that protrudes quite high above the water surface, with radio antennas installed on it.

The ORP Arctowski ship is equipped with a typical USA Sperry Marine Navigat X MK1 gyrocompass and the above-mentioned POS MV OceanMaster system. This system was considered the reference system for the surveys carried out using a Navigat X MK1 gyrocompass as well as the additionally installed, for the duration of the experiments, Navigat FOG 3000 compass, which was regarded as the basic research object, and Furuno SC50 satellite compass, which was used for comparisons. Based on the technical documentation for these instruments and after considering that the experiments were carried out at a latitude of 54.5° (Gdańsk Bay), the following accuracies should be expected: Navigat X MK1—0.7°, Furuno SC50—0.5°, and Navigat FOG 3000—0.7° at the reference course accuracy (Ocean Master) of 0.2°. It was assumed that if the errors of the instruments were considered and if the reference system was subjected to a normal distribution, then the accuracy of the reference system would have a negligible effect on the analysis of the accuracy of the other compasses installed on the ORP Arctowski warship.

As for the other platforms, the measurements used for the analyses were those taken using their standard equipment and recorded when performing their statutory measurement tasks. The Wildcat catamaran is equipped with the Trimble SPS 461 system (satellite compass), while the DriX drone is equipped with the French iXblue Phins system based on a fibre-optic gyroscope (FOG). According to the equipment manufacturers’ data, the accuracy of a course determined with iXblue Phins is 0.01° [18], while it is 0.9° when using the SPS 461 [19].

The recordings cited in this article were generally performed at a sea state of two and with winds from the direction of W or SW and a speed not exceeding 2 m/s. It should be stressed that, although these conditions should be assessed as favourable, a portion of the surveys carried out in the open sea with the ORP Arctowski warship were performed under the swell conditions, which resulted in the ship’s gentle swaying of up to five degrees. This was noticeable in some recordings, resulting in greater errors in the compasses that were additionally mounted on the ship. The velocity of all vessels during the surveys was approx. 4 m/s, while the length of the recordings under discussion, which were made on the ship, was approx. 1000 s. On the smaller vessels, the measurement sections were shorter, and consequently, their duration was shorter as well, being slightly more than 500 s for the catamaran and less than 100 s for the drone. The sampling frequencies were also different, which were determined by the technical parameters of the hydrographic survey instruments (multibeam echo sounders) used on these vessels and the required accuracies of depth measurements. For the data from the ship with a classic compass, it was 0.2 Hz, while the data from the catamaran showed a frequency of 13 Hz and the data from the drone showed a frequency of 30 Hz.

On the ship, ten recordings were made, of which three characteristic ones are presented in Figure 4. What is clearly visible are the oscillations (rather compatible in terms of frequency and phase), which form a natural phenomenon as they are the symptoms of yawing. Since all sensors were installed on the same ship, they should indicate the same values.

However, the fact that differences, especially in the amplitude values, are noticeable is intriguing. In test #3, it is noticeable that the readouts of the Navigat X gyrocompass are gradually approaching the other readouts, which can be explained by the typical behaviour of a classic gyrocompass after making an abrupt turn (in this case, by 180°). Some of these differences could also be interpreted as insufficiently accurate installation of the instruments mounted for the duration of the experiments. For example, in test #2, one can notice a systematic shift of the results from the Furuno SC50 satellite compass, while in another test, this difference is not evident. What is also noticeable is the fluctuations of the indications of the classic Navigat X MK1 gyrocompass, the Furuno SC50 satellite compass, and the Navigat FOG 3000 gyrocompass, being greater than the results obtained using the Applanix reference system. This certainly results from the higher accuracy of the reference system and the presence of a mechanical component with its own dynamics in the Navigat X MK1 compass. However, the other compasses, which have no dynamic components, also exhibit a larger oscillation amplitude; the assumption is that the changes in the ship’s course were not that large, as they are not shown by the reference system. The Furuno SC50 and the Navigat FOG 3000 compasses were installed in the ship temporarily, and their installation might have an effect on the results since both devices could not be installed in the ship’s diametral plane; the FOG compass was installed near the left side, while the satellite compass was installed several metres to the right from the diametral plane on a 3 m high mast. It is likely that these shifts and rolling on the waves could have adversely affected the results.

Figure 5 shows the differences between the values indicated by the compasses under study and the indications of the reference system for the same examples.

Based on the above diagrams showing the differences between the compasses and the reference system, it could be assumed that oscillations with a period slightly longer than 100 s were the actual changes in the ship’s course (yawing). However, there is a noticeable difference in the amplitude of these oscillations, which should be interpreted as showing the varying accuracies of these compasses. However, in the examples presented here, these values are not repeatable. In test #3, the already mentioned drift of the classic gyrocompass is highly evident, manifested by the fact that its indications are gradually approaching the indications of the other compasses. This lasts for slightly more than 20 min and is, therefore, in line with Schuler’s theory if one considers that it occurs after the ship has made a 180° turn. However, the changes in the accuracy of the FOG compass in test #1, which, after approx. 300 s, begin to deviate from the indications of the other compasses, are surprising. A similar phenomenon also emerges at the end of test #3.

The mean values of the differences between the values indicated by the reference system and the compasses under study, from all the tests, show errors of approx. 1.5°. These values actually confirm the manufacturers’ declarations regarding mean error values. It should be clearly emphasised, however, that it means that a compass does not always show the true course; the ship compasses tend to show greater deviation values than the reference system, but equally often, these deviations are not synchronised in phase. In other words, there are situations where the reference system shows movement in the opposite direction than that shown by a compass.

The analysis of the recordings presented in Figure 6 in the frequency domain shows similarities, particularly in regard to the frequency close to 0.02 Hz, which confirms the assumption that this is the yawing frequency.

Although the mean value of deviations for all three compasses in relation to the reference system for all ten recordings made on the ship is close to 1.5°, slight differences occur in the individual samples. In several tests, the satellite compass shows mean errors of up to 1.2°, while the mean errors for the Navigat X MK1 more often exceed the value of 1.5°, and even up to 1.7°.

Figure 7 shows two typical recordings made on the Wildcat catamaran, one for sailing downwind and one for sailing upwind. The mean error of maintaining on the course, which is calculated as the mean error of the course in relation to the direction of the established survey line, amounts to 1.3° upwind and 1.4° downwind.

In turn, Figure 8 shows the amplitude spectrum of these recordings made at the velocities similar to those of the ship. Both Figure 7 and Figure 8 suggest that the catamaran’s yawing is characterised by a considerably shorter period (a higher frequency). This is related to the better manoeuvrability of this vessel and the use of an automaton to control it, while it should be emphasised that the aforementioned automaton is not guided by the value of the course but by the XTE value, making turns with an angle greater than the human responses in the case of the ORP Arctowski ship. At the same time, however, it should be noted that no higher frequencies occur in the amplitude spectrum, which confirms the assumption that this satellite compass, which is stabilised with inertial components, exhibits the features of an instrument with very low error oscillation frequencies. The only issue open for discussion is whether the amplitude of the demonstrated oscillations is only the amplitude of the catamaran’s yawing or it also contains a component that is the dynamic error of the compass. An analysis of a typical hydrographic vessel’s route above the bottom also suggests that an external factor has been occurring periodically, which carries the vessel off from the planned route and necessitates strong responses, confirming the information shown symbolically in Figure 2b.

Figure 9 shows an excerpt from a USV recording. As in the case of the catamaran, what draws attention is the lower yawing frequency and the lower course oscillation amplitude. The mean error of maintaining the survey line for the drone (the mean value of the course deviations from the established survey line direction) amounts to 0.9°. Two oscillation bands with different amplitudes can be noted in both the course change graphs and the amplitude spectrum graphs. A conjecture arises that this slightly higher frequency band, which is characterised by a lower amplitude, may be a characteristic of the drone’s errors, as it should also be noted that the calculation results do not result from single surveys but partially consider previous ones (they are a result of filtering). This issue requires further research, as the reason for the lower oscillations could be the impact of the environment (e.g., waves or wind), resulting in the drone being pushed from the established survey line, while higher amplitudes may be a symptom of the drone’s directional instability. It is worth noting that a similar situation, i.e., two different oscillation bands with the higher one having a markedly lower amplitude, occurs for the catamaran and a completely different compass.

## 4. Discussion

When considering the accuracy of the compass on a hydrographic vessel and whether its parameters meet the requirements that are found on such a vessel, it is important to take into account the differences between the control strategy on a typical merchant ship and that on a hydrographic vessel. Striving to stay on the established survey line results in the possible occurrence of course changes, which may be greater and more abrupt than could be expected, assuming that a hydrographic vessel essentially moves along a section of a straight line (survey line). Furthermore, these sections are usually shorter (no more than several tens of nautical miles) than the average sections of the route covered by a typical merchant vessel, particularly in deep-sea shipping (more than a hundred nautical miles). Frequent course changes by 180°, occurring when switching from one survey line to the next one and differing by exactly 180°, must take into account the issue of compass operation stability under these conditions. These conditions are known to be unfavourable for compasses built on the basis of a mechanical gyroscope and, undoubtedly, for analytical compasses as well, i.e., those without a sensor lining up with the meridian, and the course results from the processing of data (whether from FOG sensors, RLG, or satellite measurements) perform better in this application. At the same time, there is no indication that significant inaccuracies related to altering the course appear in the information on the course readouts from the analytical compasses, which undoubtedly appears in readouts from the more traditional gyrocompasses with mechanical gyroscopes.

When assessing the accuracy of hydrographic vessel control, it must be stated that, for this issue, compass accuracy is of secondary importance. What is crucial is the vessel’s ability to maintain itself on the survey line. This cannot be regarded in the same way as the directional stability of an ordinary vessel is regarded because, in hydrographic surveys, the task of maintaining the vessel on a preset route is usually paramount. Therefore, the essential information for the helmsman or the autopilot is the deviation from the preset route (XTE). As such, the strategy of a quick return to the preset route most likely results in greater changes to the course than those observed on an average vessel. The recordings under study suggest that smaller vessels maintain themselves on the survey line better than larger vessels. This is probably due to the greater manoeuvrability of smaller vessels and lower inertia, but a shortcoming of the available recordings is that the control was carried out automatically on the smaller vessels while the ship was controlled by a human. For this reason, it is not possible to provide an explanation that is contrary to this assumption. The opinions of experienced hydrographers from the ORP Arctowski ship are not in line with this view.

The analyses of the available materials, carried out in the frequency domain, seem to suggest that a vessel’s manoeuvres can have a certain effect on the accuracy of compass indications, including compasses that are not built based on inertial components. Both the FOG compass and the satellite compass exhibited accuracies similar to the accuracy of a classic gyrocompass in the situation of any slight swaying of the vessel. It seems that this may be related to the location of their installation. The compasses described in this article were temporarily installed on the ship for the duration of the experiments, and, for reasons beyond the control of the authors, this could not be performed at optimal locations. Both the Navigat3000 compass (FOG) and the SC50 (satellite) compass were positioned away from the ship’s diametral line, and the satellite compass was installed at approx. 2 m above the top deck. Under these conditions, any roll caused additional linear movements of the sensors as a function of the distance between the ship’s pivoting point and the place where the sensor was located, and this could result in slight distortions in the readouts.

## 5. Conclusions

The FOG Navigat 3000 and SC50 compasses under consideration are instruments that fulfil the functions of a ship’s gyrocompass, yet they do not use inertial sensors. Nevertheless, they are designed to be used on ships, which means that the manufacturers have respected the international requirements that are imposed on such a case [4]. This means that their accuracies are determined, to a certain extent, by the latitude at which they are operated, and in the case in question, values exceeding 1° are to be expected. These values were confirmed in practice, but it should be emphasised that the values of the same ship’s course, which were parallelly recorded by the reference system, exhibited a smaller amplitude of oscillations. It can be concluded that the turns of the ship carrying these compasses have an effect on the value of the course indicated. In an alternative version, the reference system smooths out the actual changes to the course and does not indicate the true changes to the course, which, however, would contradict the assumption of the higher accuracy of that system.

The accuracies found for the catamaran and the drone suggest that, in both cases, better accuracies were achieved. However, in the absence of a reference system, the compass indications were compared with the direction of the entire section of the route above the seabed, so this would require further research.

In contrast to the requirements for ships, no corresponding requirements for hydrographic vessels are known since, in these cases, the requirements are formulated in terms of the accuracy of the measuring point’s geographical coordinates. In an auxiliary manner, it is expected that a survey vessel should move as close as possible to the established survey line. Consequently, rather large changes in the course, which result from the efforts to quickly return the vessel to the survey line, appear in the records of the movement of such a vessel. This procedure, however, does not result in considerable changes in the accuracy of the compasses under consideration, whose accuracy is within the specification. Although these requirements are not formulated for hydrographic vessels, and it could be expected that they will have an adverse effect on hydrographic survey accuracies, no such relationship can be indicated. With regard to hydrographic surveys, information on the course is auxiliary information for a human or an autopilot that is supposed to maintain the vessel on the survey line, and it is the XTE value that remains the primary source of information. At the same time, it appears that the location of their installation in relation to the vessel’s centre of rotation is more important for the accuracy of these compasses than the course change values and frequencies, which, for compasses with mechanical gyroscopes, are an undesirable factor.

## Figures and Tables

**Figure 1 sensors-23-01254-f001:**
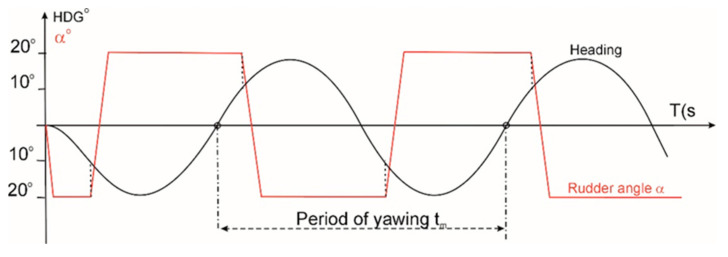
The zig-zag test (20-10 variant).

**Figure 2 sensors-23-01254-f002:**
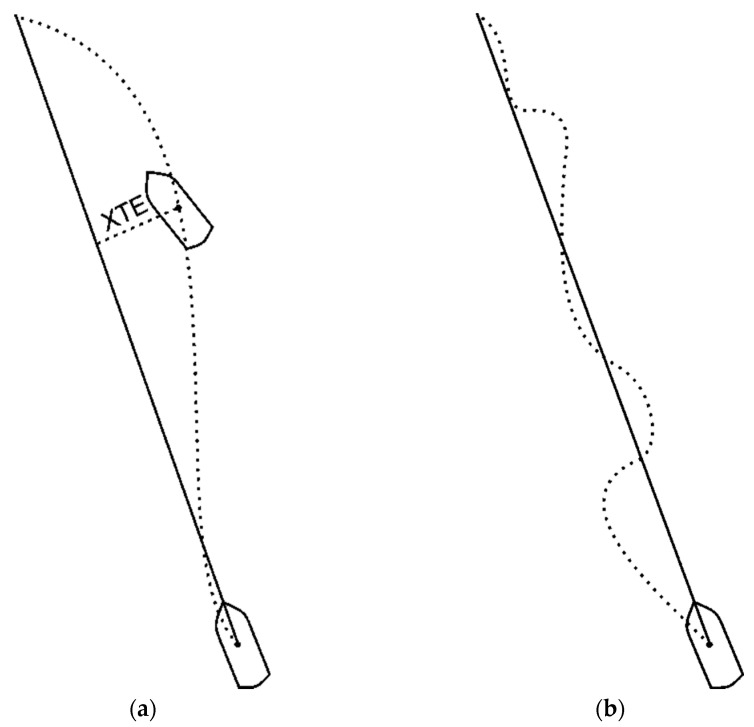
Principles of controlling the movement of a hydrographic vessel: (**a**) the value of the deviation from the planned route (Cross Track Error—XTE), and (**b**) the results in the movement of a vessel’s centre of gravity.

**Figure 3 sensors-23-01254-f003:**
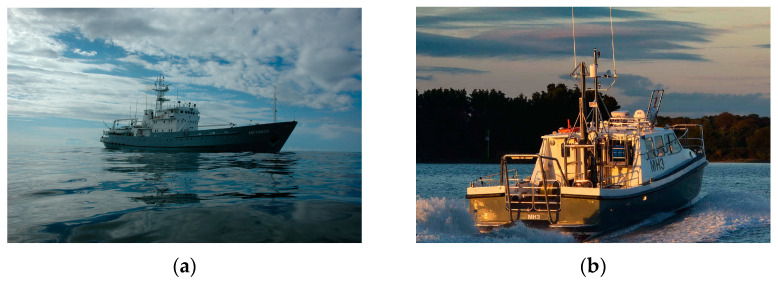
The platforms used in the study, from the left: (**a**) ORP Arctowski survey ship, (**b**) Wildcat catamaran, and (**c**) DriX drone. Sources: K. Jaskólski.

**Figure 4 sensors-23-01254-f004:**
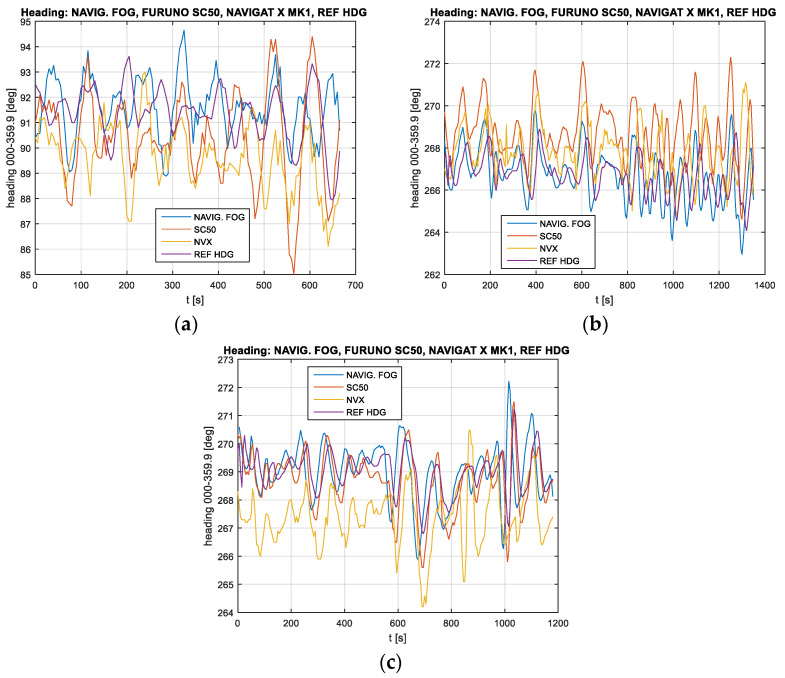
Selected recordings made on the ORP Arctowski ship: (**a**) test #1, (**b**) test #2, and (**c**) test #3.

**Figure 5 sensors-23-01254-f005:**
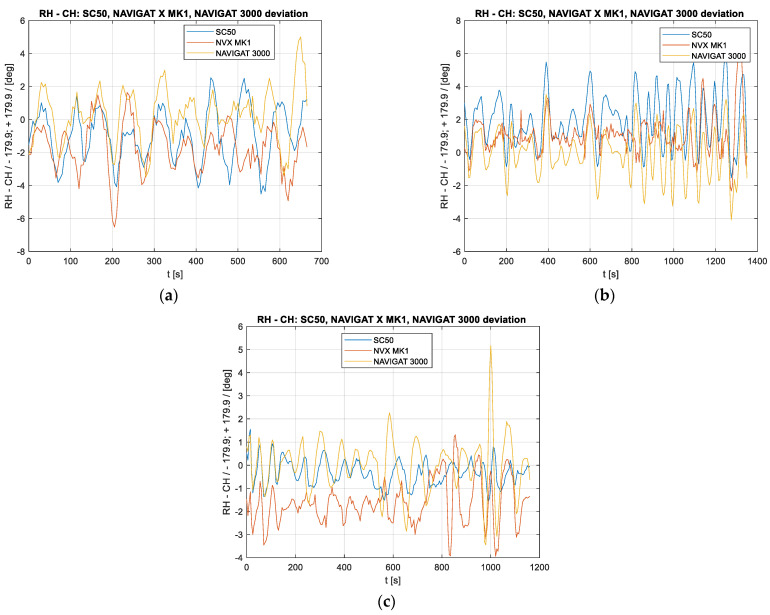
The differences between the course values indicated by the compasses under study and the reference device: (**a**) test #1, (**b**) test #2, and (**c**) test #3.

**Figure 6 sensors-23-01254-f006:**
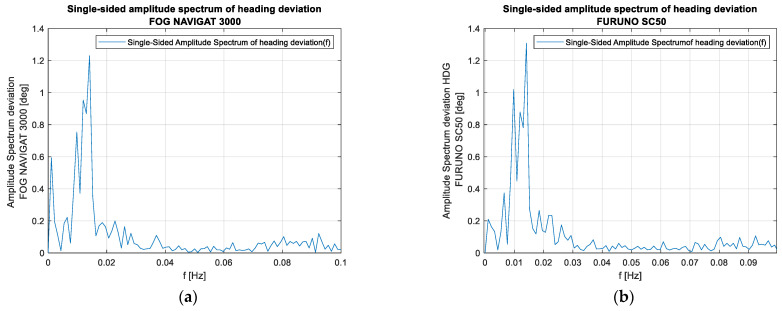
The spectrum of oscillations from recording #1 for the compasses (**a**) FOG NAVIGAT 3000 and (**b**) FURUNO SC50.

**Figure 7 sensors-23-01254-f007:**
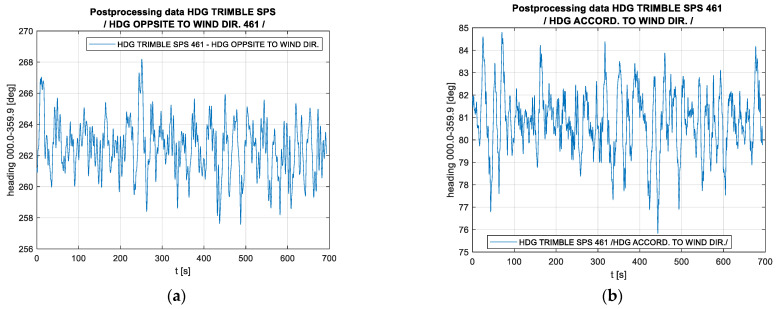
Typical recordings of the course made on the WildCat catamaran (time domain): (**a**) test #1, the vessel’s course is opposite to the wind direction, and (**b**) test #2, the vessel’s course follows the wind direction.

**Figure 8 sensors-23-01254-f008:**
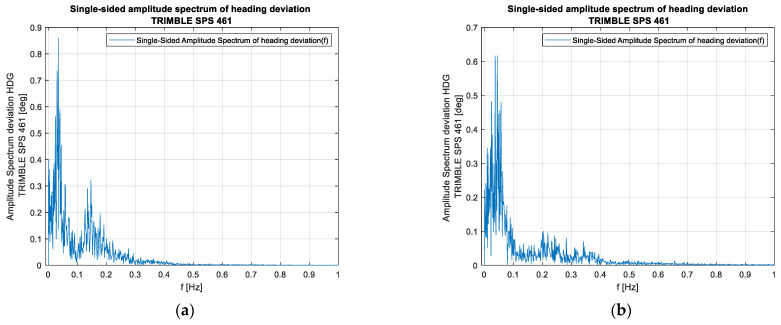
Typical recordings of the course made on the WildCat catamaran (frequency domain): (**a**) test #1, the vessel’s course is opposite to the wind direction, and (**b**) test #2, the vessel’s course follows the wind direction.

**Figure 9 sensors-23-01254-f009:**
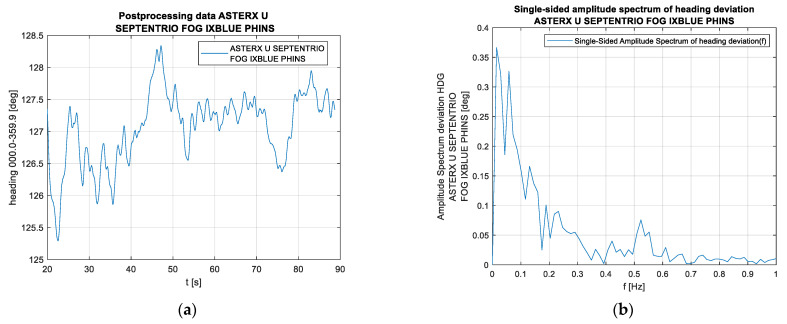
An example of a recording of the drone’s course and the amplitude spectrum of this recording: (**a**) the drone’s course in the time domain, and (**b**) the unilateral amplitude spectrum of the compass deviations in the frequency domain.

## Data Availability

Not applicable.

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
