# Peer review of "The Properties of a Ship’s Compass in the Context of Ship Manoeuvrability"

_sensors, 2023, doi:10.3390/s23031254_

Round 1

Reviewer 1 Report

The Paper presents an interesting analysis of the error frequencies of a gyroscopic compass on vessel movement. In the introduction, the authors presented the background of the work and basic concepts in marine compass navigation. In the following chapter, the authors discuss compass errors and experimental results. In the end, the authors discuss the results in conclusion. The Paper is clear, and a person with basic knowledge can understand it. The English language is understandable, and the figures are clear. While the Paper is well prepared, there are still some concerns that the authors should address, which are listed below:

1.       From the Paper, it is not clear to me why commercial compasses were used for the investigation of accuracy. Such a compass should probably go through precise certification processes where accuracy should be guaranteed.

2.       The authors claimed that it was hard to define the reference system. Why didn’t they use GPS data as a reference? Tangent on every GPS trace point should provide information about the actual direction at a specific time. GPS systems (especially on military vessels) should be very accurate, and this could be then compared to compasses. GPS location should not be so sensitive to external influences like waves therefore, it would be easier to see which part of the course error corresponds to which effect.

3.       If understood correctly, yawing is a typical response of cruise control. If the vessel's response to the rudder change is well known, why cannot this phenomenon be compensated by proper regulation of the rudder?

4.       In Figures 4, 5, and 7 it is not clear what the title on the y-axis means (000-259.9)?

5.       In Figures 8 and 9b, the frequency axis should be set to a maximum of 1 and 2 Hz

6.       In line 195, there is a typing error in a bracket

Overall, even if the Paper is well prepared, a novelty in terms of sensors is not clear to me. I believe that it would be more suitable to publish this Paper in some more marine-oriented journal where it could find a broader audience.

Author Response

Dear Sir / Madam,

Thank you for your valuable comments on the manuscript content. Your observations and comments will definitely improve the publication quality. I appreciate the effort and time spent on checking the content of the manuscript and the observations that improve the potential article quality.

 ISSUE 1: From the Paper, it is not clear to me why commercial compasses were used for the investigation of accuracy. Such a compass should probably go through precise certification processes where accuracy should be guaranteed.

ANSWER 1: Thank you for your valuable attention. Manufacturers of compass devices conduct accuracy tests under laboratory conditions. Devices rated to 0.02 deg or 0.008 deg in a laboratory environment achieve significantly lower accuracy in marine conditions. An example is the POS NAV OCEAN MASTER device. As a result, there is a belief that a ship can be steered with such accuracy. As it was emphasized in the first part of the article, the accuracy of the ship's motion is related to its inertia and the compass-automaton feedback.. The latter relationship is overlooked and its nature is very complex. Natural oscillations of the compass can be interpreted as the ship's rotation (change of course) and force unnecessary reactions to the rudder. On the other hand, the inertia of the ship's rotational movements (due to the rudder) causes the compass to deviate from its correct position. The proposed Fourier transform method aims to present the appearing deviations of the compass in the frequency domain, which gives grounds for separating the movements of the ship from the compass oscillations. The frequency of compass deviations is the basis for conducting detailed analyzes and investigating the causes of a given type of compass errors. We used the Fourier transform method to analyze the frequency of different types of compass deviations. This is particularly important in case of classic gyro compasses systematic and random deviation. When we analyze the oscillations of compass errors in the time domain and the frequency domain, we can observe different types of compass deviations appearance and frequency occurrence. In this way, it is possible to observe the moments of dynamic deviations, i.e. inertial deviations and environmental deviations, which are related to the dynamic phenomena occurring in each compass.

The study concerns three types of compasses used on vessels carrying out a hydrographic survey, fundamentally different in their dynamic properties. The reference POS NAV OCEAN MASTER was a hydrographic device. Accuracy 0.02 deg. Based on the indications of this device, the helmsman maneuvers along the hydrographic profile. In the case of other devices, a different method of determining the reference direction was proposed, i.e. determining the reference course based on the estimation of the ship's average course in the time domain. Errors resulting from the appearance of the ship's drift can be filtered out using digital signal processing methods. For example, a finite impulse response filter with a pass band or a stop band depending on the compass error frequency.

ISSUE 2:. The authors claimed that it was hard to define the reference system. Why didn’t they use GPS data as a reference? Tangent on every GPS trace point should provide information about the actual direction at a specific time. GPS systems (especially on military vessels) should be very accurate, and this could be then compared to compasses. GPS location should not be so sensitive to external influences like waves therefore, it would be easier to see which part of the course error corresponds to which effect.

ANSWER 2: I appreciate your valuable insights . We do not use a GPS device as an indication of the reference direction for the object's movement. The GPS indicates the instantaneous course over ground. Our goal is to study the heading relative to the ship's bow. Even the method of determining the heading by the satellite compass is different than the indication of  the course over ground by the GPS receiver. The solution proposed by the reviewer is perfect for road or rail traffic tests, when the indication of the course over ground and heading is almost the same. Drift and leeway appear in marine traffic.. As a result, HDG and COG may differ by several dozen degrees.

ISSUE 3: If understood correctly, yawing is a typical response of cruise control. If the vessel's response to the rudder change is well known, why cannot this phenomenon be compensated by proper regulation of the rudder?

ANSWER 3: Thank you for your valuable attention. The fundamental problem is that we do not know the exact reaction of the ship to the rudder. It depends on the vessel speed, external factors, for example the water depth, waves (especially wave height, wave period, direction of waves relative to the ship's orientation), so we are not able to completely eliminate yawing. The value of yawing in the case of ship follow-up control depends on the helmsman's experience or the quality of the autopilot and the adequacy of the settings applied to the ship's characteristics in specific weather conditions.

Thank you again for your valuable comments.
With best regards,
Authors

Reviewer 2 Report

In this paper, a survey was recorded on three platforms used for statutory hydrographic surveys for studying the properties of the ship’s compass in maneuverability. Different compass systems were used on these platforms, and the experimental results are helpful to the actual operation of the vessel. After reading the manuscript, the investigation of the experimental analysis is thorough and complete. 

Please see attached for more comments.

Author Response

Dear Sir / Madam,

Thank you for your valuable comments on the content of the manuscript. Your observations and comments will definitely improve the quality of the publication. I appreciate the effort and time spent on checking the content of the manuscript and the observations that improve the quality of the potential article

 ANSWER 1:  Thank you for your valuable attention We have supplemented the information on fiber optic gyrocompasses.

ANSWER 2: I appreciate your valuable insights . Correction has been applied to the manuscript..

ANSWER 3: Thank you for your valuable insight. kn means knots.

ANSWER 4: I appreciate your valuable insights. The designation of units has been standardized in the text of the manuscript.

ANSWER 5: Thank you for your attention. Test#3, figure 4 shows the big differences in the oscillation of the NAVIGAT X MK1 in comparison with the FOG NAVIGAT 3000 compass and the FURUNO SC50. Which is consistent with the text of the manuscript and the description of Figure 4. Test#2, figure 4 shows the systematic error of the FURUNO SC50 compass.

 ANSWER 6: : I appreciate your valuable insights . Figures 4 and 5 refer to the presentation of different values. Figure 4 shows the course variability in the time domain for 3 compasses and one reference compass. Figure 5 presents the oscillations of indications of 3 compass devices reduced by the value of the reference heading in time domain.

 ANSWER 7:  Thank you for your valuable insights. The aim of the manuscript was not to determine the best compass accuracy. The main goal was to show, that the quality (accuracy) of the compass is only one of the factors determining the accuracy of staying on the given track (measuring line).

 ANSWER 8: We greatly appreciate reviewers professional comments. In this case, the error value was determined as the heading arithmetic mean based on 30 second observations,. In fact, this confirmed the results obtained in another experiment, described in the authors' publication, which also described the method of determining the reference heading :

Jaskólski, K.; Felski, A.; Piskur, P. The Compass Error Comparison of an Onboard Standard Gyrocompass, Fiber-Optic Gyrocompass (FOG) and Satellite Compass. Sensors 2019, 19, 1942. https://doi.org/10.3390/s19081942

ANSWER 9: Thank you for your valuable attention to the measurement methodology. I think the approach is right. Unfortunately, I did not have the opportunity to mount compasses anywhere on the survey warship. The assembly locations were designated by the ship's crew. The warship performed the statutory task of hydrographic survey. With the consent of commanding officer, we installed academic equipment and made three measurements campaign. Next, we compared  outcomes with the reference heading from the hydrographic device.

Thank you again for your valuable comments.
With best regards,
Authors

Round 2

Reviewer 1 Report

The authors clarified and answered the questions according to the reviewer's comments and further clarified certain concerns regarding the paper. After another careful reading of the paper, I believe that paper can be published however, there are still several minor issues that should be addressed:

1.       In figure 4, 5 and 7, titles on the y-axis should be clarified. What do numbers like 000-259.9 means? If this is the heading direction, why are numbers on the y axis so different? This should be clarified so that is clear to the average person.

2.       In Figures 8 and 9b, the frequency axis should be set to a maximum of 1 Hz like in fig 6. This would magnify the area of interest.

3.       Reference should be added to additional text explaining the operation of the Sagnac interferometer so that readers can find a more detailed explanation if interested.

Author Response

Dear Sir / Madam,
Thank you very much for the valuable comments contained in the review. We appreciate the effort and commitment involved in working on the review.
Below we present answers to the following matters / questions.

1. In figure 4, 5 and 7, titles on the y-axis should be clarified. What do numbers like 000-259.9 means? If this is the heading direction, why are numbers on the y axis so different? This should be clarified so that is clear to the average person. 

Answer 1: Axis OY takes values from the full angle range in the circular system 000.0 - 359.9. We assumed that heading 000.0 = 360.0, hence we assumed the maximum value of 359.9 degrees and the precision of the presentation of the results on the OY axis is 0.1 deg. Figure 4 presents the dependence of heading as a function of time and, depending on the direction of the hydrographic profile, it oscillates between 270 (+/- 6 deg) and 090 (+/- 6 deg).
Figure 5, the OY axis is the heading oscillations of the three compasses minus the OCEAN MASTER reference heading. Therefore, they range from -179.9 to +179.9.
0.1 degree precision.

2. In Figures 8 and 9b, the frequency axis should be set to a maximum of 1 Hz like in fig 6. This would magnify the area of interest.

Answer 2: Thank you for your valuable attention. The Hz value on the OX axis is set automatically by Matlab according to the Nyquist frequency, which is half of the sampling frequency. As suggested, we changed the limits of the result presentation on the OX axis to 1 Hz.

3. Reference should be added to additional text explaining the operation of the Sagnac interferometer so that readers can find a more detailed explanation if interested.

Answer 3: Thank you for your valuable attention and thorough review of the content of the article. We have added a reference to the manuscript.